# GradMask: Reduce Overfitting by Regularizing Saliency

**Becks Simpson**[1]                                          becks.j.simpson@gmail.com

**Francis Dutil**[2]                                          francis.dutil@imagia.com

**Yoshua Bengio**[1]                                          yoshua.bengio@mila.quebec

**Joseph Paul Cohen**[1]                                          joseph.paul.cohen@mila.quebec

[1]*Mila, Université de Montréal*
[2]*Imagia Cybernetics*

**Editors:** Under Review for MIDL 2019

## Abstract

With too few samples or too many model parameters, overfitting can inhibit the ability to generalise predictions to new data. Within medical imaging, this can occur when features are incorrectly assigned importance such as distinct hospital specific artifacts, leading to poor performance on a new dataset from a different institution without those features, which is undesirable. Most regularization methods do not explicitly penalize the incorrect association of these features to the target class and hence fail to address this issue. We propose a regularization method, GradMask, which penalizes saliency maps inferred from the classifier gradients when they are not consistent with the lesion segmentation. This prevents non-tumor related features to contribute to the classification of unhealthy samples. We demonstrate that this method can improve test accuracy between 1-3% compared to the baseline without GradMask, showing that it has an impact on reducing overfitting.

## 1. Introduction

Overfitting can result in a model incorrectly associating some input features with a class label and being unable to unlearn this hypothesis due to an insufficient number of contradictory examples or no limit imposed by the capacity of the network (Srivastava et al., 2014). In medical imaging, small datasets are common and can come from a genuine lack of data such as imaging for rare diseases. However, in the case of datasets from combined institutions, the specific data acquisition practices of each organization can result in the presence of distinct features in the imaging which should not be used for prediction but may contribute to overfitting, such as equipment in scans or signal variations due to different acquisition parameters rather than biologic effects (Limkin et al., 2017). In fact, (Zech et al., 2018) showed that confounding factors like hospital origin could be predicted directly from imaging data, and degraded generalization performance. Therefore a method is needed to penalize models for incorrectly assigning relevance to these features that no human doctor would use for diagnosis.

Many methods have been proposed to regularize neural networks to prevent overfitting, including dropout, early stopping before validation performance worsens, introducing weight penalties such as L1 and L2 regularization and soft weight sharing (Srivastava et al., 2014; Nowlan & Hinton, 1992). Newer methods like cutout, where sections of input image are


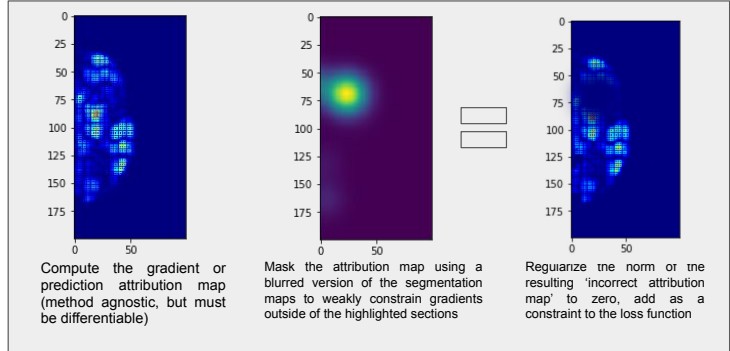

Each image has a corresponding segmentation mask

Compute the gradient or prediction attribution map (method agnostic, but must be differentiable)

Mask the attribution map using a blurred version of the segmentation maps to weakly constrain gradients outside of the highlighted sections

Regularize the norm of the resulting 'incorrect attribution map' to zero, add as a constraint to the loss function

Figure 1: A visualization of the GradMask method applied to brain tumor classification

masked out, have also shown good regularization potential for CNNs (Devries & Taylor (2017)). However, these methods generically penalize the model for capacity but without specifically addressing the need to fix incorrectly assigned feature attribution. While a model may perform better with these regularization methods and have better generalization ability, better results may be obtained by also reducing the likelihood that the learned features capture spurious correlations in the data. With modern gradient-based attribution methods, however, it is possible to highlight within an image, the features that the model considers most predictive of a particular class (Ancona et al., 2017). In a similar vein, using gradients of activations with respect to the input data had previously been applied to regularization by Rifai et al. (2011). Although the purpose was for robust feature extraction and representation construction. This study demonstrated the utility of saliency maps for some form of regularization by successfully constraining the degree to which activations can change based on the input.

Aligned with this, we propose to penalize the use of incorrect predictive features in order to make the minimum of the loss function avoid values of the parameters $\theta$ which use these incorrect features. By using input feature attribution such as computing $\frac{\partial \hat{y}_i}{\partial \mathbf{x}}$ for each input $x$ (i.e. saliency maps by Zeiler & Fergus (2014) and Simonyan et al. (2014)) we can identify where the network constructs discriminative features for a specific class $\hat{y}_i$. This representation can be regularized to penalize feature importance which is inconsistent with a given ground-truth segmentation $\mathbf{x}_{seg}$. In medical imaging for example, we penalize all features that appear outside a lesion when predicting if it is non-healthy. An example is shown in figure 1. Concretely we minimize:

$$\mathcal{L} = \sum_{\mathbf{x} \in D} \mathcal{L}_c + \left\| \frac{\partial \hat{y}_1}{\partial \mathbf{x}} \cdot (1 - \mathbf{x}_{seg}) \right\|_2, \tag{1}$$

Where $\mathcal{L}_c$ is the usual classification loss, $\hat{y}_1$ is the predicted output for the non-healthy class, and $(1 - \mathbf{x}_{seg})$ is a binary mask that covers everything outside the lesion. As an alternative to the basic saliency maps (generated per class) we also propose a 'contrast' saliency between healthy and non-healthy classes (labels $y_0$ and $y_1$ respectively). It is expected that input variance which impacts both classes is not overfitting; rather it is what

increases the distinction between the two classes that we want to regularize:

$$\mathcal{L} = \sum_{\mathbf{x} \in D} \mathcal{L}_c + \left|\left|\frac{\partial \, ||\hat{y_1} - \hat{y_0}||}{\partial \mathbf{x}} \cdot (1 - \mathbf{x}_{seg})\right|\right|_2. \tag{2}$$

## 2. Experiments

The experiments aim to demonstrate that the GradMask method decreases overfitting  measured by an improvement over the baseline of the test AUC (which typically shows overfitting if it is much lower than the train AUC). The valid AUC was used as a proxy for this in order to tune the relevant hyper-parameters of the model, per seed. Baseline experiments involved comparing the 'contrast' GradMask method to the architecture without GradMask. The number of samples for training was varied from 64 to 512 to investigate the impact on overfitting and it was found that sample sizes of 64-128 provided the first indication of improvement over the base model. Hyper-parameter search over 20 trials with a different seed each and across four datasets (three MSD and one BraTS) was performed to select the runs with the best valid AUC.

Results of these experiments, shown in Table 1, demonstrate that the gradmask method achieves between 1-4% increase in test accuracy.

| Dataset | GradMask Variant | Test AUC Mean + SD | # Samples |
|---|---|---|---|
| Liver Seg (MSD) | -None- | $0.809\pm 0.042$ | 128 |
| (Kumar & Greiner, 2019) | Contrast | **$0.836\pm0.017$** | 128 |
| Pancreas Tumor (MSD) | -None- | $0.776\pm0.019$ | 128 |
| (Kumar & Greiner, 2019) | Contrast | **$0.783\pm0.018$** | 128 |
| Brain Tumor (BraTS) | -None- | **$0.826\pm0.026$** | 128 |
| (Menze et al., 2015) | Contrast | $0.798\pm0.019$ | 128 |
| Cardiac Seg (MSD) | -None- | $0.864\pm0.032$ | 64 |
| (Kumar & Greiner, 2019) | Contrast | **$0.877\pm0.031$** | 64 |

Table 1: Mean Test AUC and Standard Deviation over 20 seeds for GradMask - Contrast compared to no GradMask (-None-)

## 3. Conclusion

The results demonstrate that this method is able to decrease overfitting on small datasets. Limitations of the experiments were that on some tasks, the CNN model used could not achieve suitable baseline accuracy meaning that improvements from GradMask were also hindered. Potentially image size played a role in this due to the small tumor size and resizing the image slices may have eroded the tumors to a point that made the tasks effectively impossible for some examples. This will be addressed in future work along with investigation of additional methods of gradient attribution such as DeepLIFT and the performance of GradMask with reduced number of available segmentation masks during training.

## Acknowledgments

This work is partially funded by a grant from the Fonds de Recherche en Sante du Quebec and the Institut de valorisation des donnees (IVADO). This work utilized the supercomputing facilities managed by Mila, NSERC, Compute Canada, and Calcul Quebec. We also thank NVIDIA for donating a DGX-1 computer used in this work.

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
