# OpenReview forum: "GradMask: Reduce Overfitting by Regularizing Saliency"
_MIDL.io/2019/Conference/Abstract — MIDL Abstract 2019_

### Official Review · AnonReviewer1 · 2019-04-29
**Novel regularization technique that discourages spurious feature attribution**

**Rating:** 4
**Confidence:** 2

**Review:**

Pros:
Nice methodological contribution.
Unlike regularization techniques that limit the capacity of the model, the authors propose a method to reduce overfitting by discouraging incorrect feature attributions. This is done by constraining the saliency maps to contain only the correct features. This requires a prior knowledge of what constitutes as 'correct features', of course. But, in the presence of such prior knowledge, the proposed method provides a neat framework for incorporating the same.

Cons:
Experiments have not been clearly described.
 * It is unclear to me as to what is the network trained to do - segmentation or classification as healthy or diseased? I assume that it was trained for classification and slices with only background segmentation annotations were labelled as 'healthy' and the rest as 'diseased'. Is this correct?
 * What happens when GradMask is used with per-class saliency maps as against contrast saliency maps?
 * Why are 3 results shown with 128 training samples and the 4th result with 64 training samples?

I recommend acceptance on the basis of methodological contribution, but would encourage the authors to improve the experiments in a following full paper submission, if one is planned.

---

### Official Review · AnonReviewer2 · 2019-05-01
**Dubious method in a unrealistic setting.**

**Rating:** 1
**Confidence:** 3

**Review:**

This paper tries to tackle the problem of limited available labeled data for classification, in several medical challenges. The proposed solution is therefore to use segmentation masks to improve the results.

The paper is badly written, and keep switching between different goals: dealing with small datasets, reducing overfitting, preventing the network to learn features specific to the hospital origin. At the end of the paper, the only demonstrated hypothesis is that using more data improves AUROC. The claim that the method decreases overfitting in uncorroborated by the showed data.

The main intuition of the paper is that a CNN will learn features specific to the hospital origin, therefore reducing generalization performances with new data from other hospitals. To prevent that, the authors propose to use saliency maps to see which pixels the CNN uses to make its decision. Their idea is to allow the network to discriminate only from pixels within the object, and not outside of it, which is a massive boost in supervision.
Yet, the authors fail to show or explain that hospital specific features are contained only in the background area of the image.

If segmentation masks are available, why not do image segmentation ? Especially since doctors are often more interested in a segmentation mask than a binary label for the whole image.
Or, conversely, if there is resources to manually segment the images, why not instead annotate more of them ?

---

### Decision · Program_Chairs · 2019-05-06
**Acceptance Decision**

Accept